# Positive Mental Health in University Students and Its Relations with Psychological Vulnerability, Mental Health Literacy, and Sociodemographic Characteristics: A Descriptive Correlational Study

**DOI:** 10.3390/ijerph19063185

**Published:** 2022-03-08

**Authors:** Sónia Teixeira, Carme Ferré-Grau, Teresa Lluch Canut, Regina Pires, José Carlos Carvalho, Isilda Ribeiro, Carolina Sequeira, Teresa Rodrigues, Francisco Sampaio, Tiago Costa, Carlos Alberto Sequeira

**Affiliations:** 1CINTESIS—Center for Health Technology and Services Research, Porto School of Nursing, 4200-072 Porto, Portugal; soniateixeira.esenf@gmail.com (S.T.); regina@esenf.pt (R.P.); zecarlos@esenf.pt (J.C.C.); isilda.ribeiro@esenf.pt (I.R.); teresarodrigues@esenf.pt (T.R.); 2Faculty of Nursing, Universitat Rovira y Virgili, 43003 Tarragona, Spain; carme.ferre@urv.cat; 3Department of Public Health, Mental Health and Maternal and Child Health Nursing, Nursing School, University of Barcelona, 08007 Barcelona, Spain; tlluch@ub.edu; 4Faculty of Medicine, University of Coimbra, 3004-531 Coimbra, Portugal; carolinasequeira7@gmail.com; 5CINTESIS—Center for Health Technology and Services Research, Higher School of Health Fernando Pessoa, 4200-253 Porto, Portugal; fsampaio@ufp.edu.pt; 6CINTESIS—Center for Health Technology and Services Research, Centro Hospitalar de Vila Nova de Gaia/Espinho, 4434-502 Vila Nova de Gaia, Portugal; tiagofilipeoliveiracosta@gmail.com

**Keywords:** mental health, mental health literacy, psychological vulnerability, health promotion, adult

## Abstract

This study aimed to evaluate positive mental health (PMH) and its relation with sociodemographic characteristics, mental health literacy, and the psychological vulnerability scale (PVS) in Portuguese university students aged 17 to 62. A descriptive correlational study was carried out. An online survey was conducted to evaluate demographic variables, and several questionnaires were applied to evaluate positive mental health, psychological vulnerability, and mental health literacy. The data was collected from 1 November 2019 to 1 September 2020. Overall, 3405 students participated in the study. The results show that 67.8% of students revealed a high level of PMH, 31.6% presented a medium level of PMH, and 0.6% had a low level of PMH. Male students reported higher personal satisfaction (*t* (3170) = −2.39, *p* = 0.017) and autonomy (*t* (3170) = −3.33, *p* = 0.001) in PMH compared to female students. Students without a scholarship scored higher (*t* (3127) = −2.04, *p* = 0.42) in PMH than students who held a scholarship. Students who were not displaced from their home reported higher (*t* (3170) = −1.99, *p* = 0.047) self-control in PMH than those displaced from their home. University students with higher PMH results had lower PVS results and higher literacy results. The findings of this study will contribute to identifying students’ PMH intervention needs.

## 1. Introduction

The COVID-19 pandemic has reinforced the notion that mental health is as important as physical health and that both are often empirically related. Therefore, countries worldwide should disclose factual data and information on mental health to politicians and the general public, allowing the implementation of measures that promote positive mental health and contribute to lowering the rates of mental disorders and subsequent social burdens [1].

Despite not being widely adopted, the term positive mental health was first designated by Marie Jahoda in 1958 [2]. It considers health promotion as an important aspect of society and the dynamic life cycle of its citizens; that is, society needs to keep healthy and optimize what is already good. Thus, positive mental health represents a specific instance resulting from the interaction of various factors within mental health and is based on the strengthening and development of the optimal functioning of the human being [3,4]. The Multifactorial Model of Positive Mental Health is based on six factors: Personal Satisfaction (F1), Prosocial Attitude (F2), Self-control (F3), Autonomy (F4), Problem-solving and Self-actualization (F5), and Interpersonal Relationship Skills (F6) [4].

The pandemic reinforced the idea that mental health is as or more important than physical health and that they are most often interlinked.

Several studies have focused on mental health, but few have addressed positive mental health using the Multifactorial Model. For example, some studies were focused on assessing the positive mental health of black college students [5]. Other studies have focused on the application of the Positive Mental Health Questionnaire to university professors from different nursing schools in Catalunya [6], health professionals working in mental health services [7], and adults with chronic physical health problems recruited from a primary care centre in the province of Barcelona [8]. The prevalence of positive mental health in Chinese adolescents has been explored and reported as higher than the results found in most previous studies [9]. Moreover, the results suggest that Chinese adolescents with favourable socio-economic situations, lifestyle, social support, and school life tend to experience better positive mental health than those not having this type of background [9].

Furthermore, studies investigating the frequency of positive mental health and its correlates are still uncommon compared to studies on mental disorders. Interest in assessing, for example, students’ positive mental health is increasing [9], but research is still scarce, and few studies have discussed the levels of positive mental health in adult students following the Multifactorial Model of Positive Mental Health. In countries such as Portugal, no studies have been found interpreting the prevalence of positive mental health in students and its relationship with other variables such as scholarship, place of habitual residence, psychological vulnerability, and mental health literacy. Nonetheless, this type of study helps create intervention guidelines to promote mental health in schools, universities, and communities.

The term mental health literacy is defined as the knowledge and beliefs about mental disorders that enable a person to recognise, manage, or prevent a mental health problem. Due to its importance, mental health literacy interventions have received increasing attention as a strategy to promote positive mental health, improve the early identification of disorders, reduce stigma, and improve help-seeking behaviours [10].

Current research with higher education students shows that psychological vulnerability is negatively correlated with adaptive constructs and positively associated with negative health outcomes [11]. Psychological vulnerability refers to a cognitive structure that makes individuals more fragile under stressful environments, assuming that some people are more affected by stressful events than others [11]. 

University students are exposed to a group of variables that cause stress. Therefore, it is important to assess students’ positive mental health, mental literacy, and vulnerabilities to intervene and develop and increase or maintain their mental health capacity [11].

The main objective of this study was to assess students’ positive mental health using the Positive Mental Health Questionnaire [4]. In addition, this study sought to evaluate the relationship between positive mental health and sociodemographic characteristics, mental health literacy, and psychological vulnerability in Portuguese students.

Our hypotheses were:

**Hypothesis** **1** **(H1).**
*Positive mental health is positively associated with mental health literacy.*


**Hypothesis** **2** **(H2).**
*Positive mental health is negatively associated with psychological vulnerability.*


**Hypothesis** **3** **(H3).**
*Mental health literacy is negatively associated with psychological vulnerability.*


**Hypothesis** **4** **(H4).**
*There are significant differences between men and women in the level of positive mental health.*


**Hypothesis** **5** **(H5).**
*There are significant differences between having and not having a scholarship in the level of positive mental health.*


**Hypothesis** **6** **(H6).**
*There are significant differences between being and not being displaced from home in the level of positive mental health.*


## 2. Materials and Methods

### 2.1. Study Design

A descriptive correlational study was conducted and reported following the STROBE statement; that is, following the Strengthening the Reporting of Observational Studies in Epidemiology checklist, used in articles reporting observational research.

### 2.2. Setting

The research was carried out using online questionnaires applied via institutional email using a link sent to each school/university student. Data were collected between 1 November 2019 and 1 September 2020.

### 2.3. Participants

Participants needed to meet the following criteria to be included in the study: (1) be enrolled in a higher education programme; (2) be ≥17 years old; (3) not have a diagnosis of psychiatric pathology (this information was asked in the sociodemographic questionnaire); and (4) agree to participate in the study by signing an online informed consent. Participants who did not have a good command of Portuguese were excluded. 

Overall, 3405 questionnaires were completed. Of those, 6.3% were excluded because some items’ information was incomplete.

The final sample consisted of 3189 students from various higher education institutions studying health, education, arts, tourism, architecture, and communication programmes in mainland Portugal and the islands.

### 2.4. Data Sources/Measurement

The students participating in the study were evaluated at a single time-point using a sociodemographic questionnaire and psychometric instruments validated for the Portuguese population.

Sociodemographic data (gender, age, marital status, parents’ educational level, enrolment year, displacement from the students’ home to go to university, and a yes/no question about the students receiving a scholarship) were collected and used to characterize the sample. In this regard, it is worth noting that scholarships are granted only to low-income university students.

The primary outcome measured in this study was positive mental health, assessed with the Positive Mental Health Questionnaire (PMHQ). The PMHQ measures positive global mental health by evaluating six factors: personal satisfaction, prosocial attitude, self-control, autonomy, problem-solving and self-actualization, and interpersonal relationship skills. The secondary outcomes assessed in this study were psychological vulnerability, measured with the Psychological Vulnerability Scale (PVS), and mental health literacy, measured with the Mental Health Knowledge Questionnaire (MHKQ). The psychometric instruments are detailed below.

#### 2.4.1. PMHQ

The PMHQ consists of 39 items [4]. The scores in this scale go from 39 (minimum score) to 156 (maximum score). The higher the score obtained, the higher the positive mental health level. Therefore, we can categorize different levels of positive mental health as low or languishing (results from 39 to 78), intermediate (results from 79 to 117), and high or flourishing (results from 118 to 156) [12]. The responses to each item are presented on a 4-point Likert scale (1 to 4) concerning the frequency of each individual’s life statement. Thus, each person must respond according to the frequency that best characterizes his/her case: “Always or almost always,” “Most of the time,” “Sometimes,” and “Rarely or never.” Of the 39 items, 19 appear positively formulated, and 20 are negatively formulated. The items positively stated are 4, 5, 11, 15–18, 20–23, 25–29, 32, and 35–37. Therefore, according to the wording of the item (positive or negative statement), the answers presented on the Likert scale will have different scores or values [4,13]. 

The PMHQ has proven to be an instrument with very good internal consistency (Cronbach α = 0.92), with each of the factors having an internal consistency ranging from 0.60 to 0.84. In addition, the instrument also presents a good test–retest reliability of −0.98 [13].

#### 2.4.2. PVS

The psychological vulnerability scale developed in the United States was designed to identify vulnerable individuals in adult, chronically ill populations. However, in Europe, studies were carried out in general populations revealing a good internal consistency [11]. The PVS is a self-administered instrument to obtain information about the psychological vulnerability of an individual. It is designed to screen for maladaptive cognitive patterns, such as dependence, perfectionism, a need for external sources of approval, and generalized negative attributions. The PVS is a six-item scale. Each item is measured in a 5-point Likert frequency scale (1–5), where a higher score corresponds to a greater frequency (does not describe me at all = 1 to describes me very well = 5). Possible total scores range from 6 to 30, with higher scores indicating a greater psychological vulnerability. In the original version, Cronbach’s α coefficient ranged from 0.71 to 0.87. In Portugal, the reliability of this scale was verified through the assessment of internal consistency, evidencing positive outcomes (Cronbach’s α = 0.73) in a sample of higher education students [11].

#### 2.4.3. MHKQ

The MHKQ questionnaire was developed by Chaves, Sequeira, and Duarte & Gonçalves [10] using a 5-point Likert scale. Higher scores correspond to a greater frequency (strongly disagree = 1 to strongly agree = 5). It is divided into three parts. The first part has 16 items that assess mental health knowledge. The second part consists of 4 items that evaluate mental health promotion activities. Finally, the third part has 10 items that assess what is important for good mental health (resources).

### 2.5. Data Analysis

The SPSS^®^ software version 24.0 was used for statistical analysis. Descriptive statistics were employed for the students’ characterization. Absolute and relative frequencies were used to describe qualitative variables, and the mean and standard deviation (SD) were used to describe quantitative variables. The independent samples *t*-test was used to assess the differences between the variables’ gender, scholarship, and displacement from the habitual residence in relation to positive mental health. Pearson’s correlation coefficient was used to determine the correlation between the PMHQ, the PVS, and the MHKQ. A significance level of 0.05 was considered. 

### 2.6. Ethical Considerations

The study was approved by the ethics committee and authorized by the executive committees of the institutions involved. Informed consent was obtained from all participants. All information provided by the participants was collected in a de-identifiable form.

## 3. Results

The sample was composed of 3189 participants, 2503 female students, and 669 (21.1%) male students. Table 1 shows the predominance of female students (78.9%). The average age of the students was 24 years, with a minimum age of 17 years, maximum age of 62 years, and a standard deviation of 7 years. Most of the students were single (89.6%), enrolled in the first year of their programme of studies (31.2%), living in their habitual home (57.7%), and 62.4% did not hold any scholarship. Regarding the parents’ schooling, most of the students had a father with a basic level of education—grades 5 to 9 (36.5%), and a mother with a higher academic level (33.0%).

As shown in Table 2, positive mental health was good in 67.8% (scores 118 to 156), moderate in 31.6% (scores 79 to 117), and low in 0.6% (scores 39 to 78) of the students.

**Hypothesis** **1** **(H1).**
*Positive mental health is positively associated with mental health literacy.*


As shown in Table 3, we found a weak positive association between the global score of positive mental health and literacy—mental health knowledge r = 0.101, *p* < 0.01, and literacy—mental health resources r = 0.150, *p* < 0.01. The association among the factors goes from strong to weak.

**Hypothesis** **2** **(H2).**
*Positive mental health is negatively associated with psychological vulnerability.*


Observing Table 4, we observed a moderate negative correlation between the PMH global and PVS (r = −0.629, *p* < 0.01). Therefore, university students with high PMH results have low PVS results.

**Hypothesis** **3** **(H3).**
*Mental health literacy is negatively associated with psychological vulnerability.*


A weak positive correlation was found between mental health literacy (mental health knowledge) and psychological vulnerability (r = 0.037, *p* < 0.05), meaning that students with higher mental health literacy knowledge had a higher psychological vulnerability.

We found a moderate negative correlation between literacy–mental health resources and PVS (r = −0.063, *p* < 0.01) (Table 5).

The relationships between levels of positive mental health (good, moderate, and low) and gender of university students, displacement from home, and holding a scholarship are explored in Table 6.

**Hypothesis** **4** **(H4).**
*There are significant differences between men and women in the level of positive mental health.*


According to the analysis shown in Table 7, statistically significant differences were found between students’ gender and the score in the “Personal Satisfaction” factor of the PMHQ (t (3170) = −2.39, *p* = 0.017). On average, male students reported higher personal satisfaction (M = 3.2685; SD = 0.61073) compared to female students (M = 3.2065; SD = 0.59370).

Furthermore, statistically significant differences were found between students’ gender and scores in the “Autonomy” factor of the PMHQ (t (3170) = −3.33, *p* = 0.001). On average, men reported greater autonomy (M = 3.1733; SD = 0.59978) compared to women (M = 3.0883; SD = 0.58132). Regarding the global score of PMH, no statistically significant differences were found between men and women.

**Hypothesis** **5** **(H5).**
*There are significant differences between having and not having a scholarship in the level of positive mental health.*


Students without a scholarship (M = 3.1862; SD = 0.38834) scored significantly higher (t (3127) = −2.04, *p* = 0.42) in positive mental health than students who held a scholarship (M = 3.1564; SD = 0.40167). Moreover, students without a scholarship scored significantly higher for the “Self-Control” factor (t (3127) = −2.92, *p* = 0.004), “Autonomy” factor (t (3127) = −2.30, *p* = 0.02), and “Interpersonal Relationship Skills” factor (t (3127) = −2.53, *p* = 0.01), compared to students who held a scholarship (Table 8).

**Hypothesis** **6** **(H6).**
*There are significant differences between being and not being displaced from home in the level of positive mental health.*


The students who were not displaced from their habitual home (M = 2.7995; SD = 0.63635) reported significantly higher self-control than those displaced from their home (t (3170) = −1.99, p = 0.047) (Table 9). Regarding the global PMH score, no statistically significant differences were found between university students displaced from their homes and those who were not.

## 4. Discussion

Several notable findings of positive mental health in Portuguese students were obtained with this research.

Most of the students showed high levels of positive mental health. Our results are in agreement with a study conducted with Portuguese and Spanish nursing students, in which 41.1% of the participants reported a good level of PMH, 58.4% a moderate level, and only 0.5% indicated a low level [14]. Young people with higher levels of PMH tend to reach higher education levels and express a greater competence at the adult age [15]. Assessing and monitoring students’ positive mental health is becoming increasingly important, as the evidence shows that positive mental health complements mental disorder screening in mental health surveillance, the prediction of suicidal behaviour, and the impairment of academic performance [16]. In addition, other studies with medical students suggest that positive mental health also attenuates some of the adverse consequences of burnout [17].

University students with higher scores of PMH also scored higher in mental health literacy. These data are in accordance with previous studies [15,16,18,19] that linked knowledge to the possibility of taking action to benefit one’s health and the health of other persons, which, in turn, showed that mental health literacy has several components [10]. As stated in the definition of positive mental health, this possibility of knowledge leads to action and enhances a person’s capacity to optimize PMH factors that require more attention and/or maintain optimum levels of PMH [1].

Negative associations between PMH and PMH factors with PVS were also found. Students with good PMH have a low psychological vulnerability. This association may indicate that having good positive mental health and being aware of their vulnerability can discriminate the interpretation of students’ psychological vulnerability in clinical samples [11].

The students who reported more knowledge about PMH were more psychologically vulnerable. This association does not yet seem to have evidence to support it. Therefore, we recommend further investigation to confirm the association and to justify it. We believe that mental health knowledge can be reflected in recognizing factors that influence mental health and the prevalence of problems. In turn, this recognition may have led to a perception of fragility and risk of experiencing mental health problems in students. In addition, greater knowledge about resources to mobilize when addressing mental health is associated with a lower psychological vulnerability. Therefore, university students with greater knowledge about mental health are more prepared to deal with transitions throughout the life cycle, which allows them not to become so psychologically vulnerable [20].

Our findings showed no significant differences in the levels of the “PMH Global” between men and women. However, most of the students with a low level of PMH were female. These findings are in line with existing research [14]. Notwithstanding, the significant differences in personal satisfaction and autonomy between male and female students should be further explored. Male students reported greater personal satisfaction and autonomy than females. Women have been described in the literature as having less access to power than men, decision-making positions, work opportunities, and leisure time [21]. Therefore, this may be reflected in the lower perceived autonomy. Despite less favourable objective conditions for women worldwide, in other studies, women were more satisfied with their lives [22].

Students without a scholarship reported greater PMH, self-control, autonomy, and interpersonal relationship skills. This finding demonstrates that good social support is a protective and essential factor in enhancing the positive mental health of university students [18].

Students living in their homes reported higher levels of PMH, specifically self-control. In addition, students who continue residing in their homes while attending university are familiar with the resources in their area of residence and have access to family support. These factors can help minimize conflict and stress situations. On the other hand, students living in a new city may not know the resources available and lack an effective family support network. Other studies suggest that emotional stability is a constant and positive aspect of personality in predicting happiness among young people [23].

Although we achieved high levels of “PMH Global”, further intervention in vulnerable groups should be carefully considered. In this way, pathological outbreaks can be avoided when people experience ineffective transitions or daily setbacks. Programmes that promote PMH can be found in the literature [1], and free smartphone apps that improve PMH levels already exist [19]. In addition, schools are privileged environments to equip students with strategies to promote positive mental health [24].

### Limitations

Replicating this study with undifferentiated working groups and within different contexts in Portugal and the European community would allow the generalisability of the results.

One of the limitations identified concerns the type of study carried out—cross-sectional, which hinders determining the cause-and-effect relationships between the variables. Moreover, since the research involved the participants’ self-report, the risk of response bias should be considered. In addition, the wide age range of the sample can influence the interpretation of the results since university students in their twenties have different family and social and cultural experiences than, for instance, university students in their sixties.

Finally, part of our data collection took place during the COVID-19 pandemic. Thus, our study included pre-pandemic data and data collected after the beginning of the COVID-19 pandemic, which may have influenced some of our findings.

## 5. Conclusions

Our results are in line with previous studies that suggest the need to invest in strategies to promote university students’ mental health by providing training and enabling them to balance work situations, and raising awareness about the importance of care.

We show that it is necessary to intervene in the prevention and promotion of positive mental health by implementing properly validated and structured programmes. Thus, the implementation and evaluation of mental health promotion strategies and practices are essential tools to promote and support action for positive mental health.

Moreover, our study identified vulnerable groups that need intervention and attention from health professionals, namely mental health nurses, psychologists, psychiatrists, and other mental health professionals.

## Figures and Tables

**Table 1 ijerph-19-03185-t001:** Students’ characterization.

	n	%	Mean	Median	Min	Max	SD
Gender							
Male	669	21.1					
Female	2503	78.9					
Age	3189		23	24.00	17	62	7
Marital Status							
Single	2843	89.6					
Married/Cohabiting	291	9.2					
Divorced	34	1.1					
Widow(er)	5	0.2					
Father’s academic background							
Basic education:							
1st cycle (grades 1 to 4)	563	1.1					
2nd and 3rd cycles (grades 5 to 9)	1154	36.5					
Upper secondary education	964	30.5					
Bachelor’s degree	362	11.5					
Master’s degree	91	2.9					
Doctorate	27	0.9					
Mother’s academic background							
Basic education:							
1st cycle (grades 1 to 4)	429	13.5					
2nd and 3rd cycles (grades 5 to 9)	1003	31.6					
Upper secondary education	1048	33.0					
Bachelor’s degree	547	17.2					
Master’s degree	117	3.7					
Doctorate							
Enrolment year							
1st year	977	31.2					
2nd year	782	25.0					
3rd year	740	23.6					
4th year	520	16.6					
5th year	74	2.4					
6th year	38	1.2					
Is the student displaced from his/her home to go university?							
Yes	1342	42.3					
No	1830	57.7					
Does the student hold a scholarship?							
Yes	1177	37.6					
No	1952	62.4					

**Table 2 ijerph-19-03185-t002:** Levels of positive mental health.

Positive Mental Health (n = 3405)	N	%
Good (118–156)	2307	67.8
Moderate (79–117)	1077	31.6
Low (39–78)	21	0.6

**Table 3 ijerph-19-03185-t003:** Correlation between PMH and MHKQ.

	F1 PMH	F2 PMH	F3 MH	F4 PMH	F5 PMH	F6 PMH	PMH Global	Literacy–Mental Health Knowledge	Literacy–Mental Health Activities	Literacy–Mental Health Resources
F1 PMH	1	0.222 **	0.586 **	0.594 **	0.556 **	0.446 **	0.827 **	0.033	−0.016	0.092 **
F2 PMH		1	0.247 **	0.195 **	0.414 **	0.501 **	0.506 **	0.176 **	−0.186	0.142 **
F3 PMH			1	0.459 **	0.614 **	0.372 **	0.762 **	0.012	0.002	0.066 **
F4 PMH				1	0.465 **	0.358 **	0.703 **	0.005	−0.014	0.082 **
F5 PMH					1	0.495 **	0.832 **	0.140 **	−0.097	0.141 **
F6 PMH						1	0.704 **	0.126 **	−0.096	0.156 **
PMH Global							1	0.101 **	−0.078	0.150 **
Literacy–mental health knowledge								1	−0.417	0.104 **
Literacy–mental health activities									1	−0.072 **
Literacy–mental health resources										1

** Correlation significant *p* < 0.01 (2-tailed). * Correlation significant *p* < 0.05 (2-tailed).

**Table 4 ijerph-19-03185-t004:** Correlation between PMH and PVS.

	F1 PMH	F2 PMH	F3 PMH	F4 PMH	F5 PMH	F6 PMH	PMH Global	PVS
F1 PMH	1	0.222 *	0.586 *	0.594 *	0.556 *	0.446 *	0.827 *	−0.637 *
F2 PMH		1	0.247 *	0.195 *	0.414 *	0.501 *	0.506 *	−0.120 *
F3 PMH			1	0.459 *	0.614 *	0.372 *	0.762 *	−0.549 *
F4 PMH				1	0.465 *	0.358 *	0.703 *	−0.562 *
F5 PMH					1	0.495 *	0.832 *	−0.464 *
F6 PMH						1	0.704 *	−0.294 *
PMH Global							1	−0.629 *
PVS								1

* Correlation significant *p* < 0.01 (2-tailed).

**Table 5 ijerph-19-03185-t005:** Correlation between MHKQ and PVS.

	Literacy–Mental Health Knowledge	Literacy–Mental Health Activities	Literacy–Mental Health Resources	PVS
Literacy–mental health knowledge	1	−0.417 **	0.104 **	0.037 *
Literacy–mental health activities		1	0.247 **	0.018
Literacy–mental health resources			1	−0.063 **
PVS				1

** Correlation significant *p* < 0.01 (2-tailed). * Correlation significant *p* < 0.05 (2-tailed).

**Table 6 ijerph-19-03185-t006:** Positive mental health levels with variables such as gender, home, and scholarship.

Positive Mental Health (n = 3390)	Male, n (%)	Female, n (%)
Good	498 (69.7)	1796 (67.1)
Moderate	214 (30.0)	861 (32.2)
Low	2 (0.3)	19 (0.7)
Positive Mental Health (n = 3390)	Living out of home to go to university, n (%)	Living in the habitual home while going to university, n (%)
Good	960 (66.6)	1336 (68.6)
Moderate	471 (32.7)	602 (30.9)
Low	11 (0.8)	10 (0.5)
Positive Mental Health (n = 3341)	With a scholarship, n (%)	Without a scholarship, n (%)
Good	836 (66.3)	1425 (68.5)
Moderate	419 (33.2)	640 (30.8)
Low	6 (0.5)	15 (0.7)

**Table 7 ijerph-19-03185-t007:** Student’s *t*-test to assess gender and PMH.

Variable	Gender	n	M	SD	t (df) p
Factor 1—Personal Satisfaction	Female	2503	3.21	0.59	−2.39 (3170) 0.017
	Male	669	3.27	0.61	
Factor 2—Prosocial Attitude	Female	2503	3.61	0.35	7.86 (3170) 0.000
	Male	669	3.48	0.41	
Factor 3—Self-control	Female	2503	2.72	0.63	−10.85 (3170) 0.000
	Male	669	3.01	0.59	
Factor 4—Autonomy	Female	2503	3.09	0.58	−3.33 (3170) 0.001
	Male	669	3.17	0.59	
Factor 5—Problem-solving and Self-actualization	Female	2503	3.15	0.48	−0.35 (3170) 0.724
	Male	669	3.16	0.49	
Factor 6—Interpersonal Relationship Skills	Female	2503	3.21	0.48	3.60 (3170) 0.000
	Male	669	3.13	0.49	
PMH Global	Female	2503	3.17	0.39	−1.95 (3170) 0.052
	Male	669	3.20	0.39	

**Table 8 ijerph-19-03185-t008:** Student’s *t*-test for scholarship and PMH.

Variable	Scholarship	n	M	SD	t (df) p
Factor 1—Personal Satisfaction	Yes	1177	3.21	0.58	−7.34 (3127) 0.463
	No	1952	3.23	0.61	
Factor 2—Prosocial Attitude	Yes	1177	3.57	0.38	−1.15 (3127) 0.250
	No	1952	3.58	0.37	
Factor 3—Self-control	Yes	1177	2.74	0.62	−2.92 (3127) 0.004
	No	1952	2.81	0.65	
Factor 4—Autonomy	Yes	1177	3.07	0.59	−2.30 (3127) 0.022
	No	1952	3.12	0.59	
Factor 5—Problem-solving and Self-actualization	Yes	1177	3.15	0.49	−0.27 (3127) 0.788
	No	1952	3.16	0.49	
Factor 6—Interpersonal Relationship Skills	Yes	1177	3.16	0.48	−2.53 (3127) 0.012
	No	1952	3.21	0.49	
PMH Global	Yes	1177	3.16	0.39	−2.04 (3127) 0.042
	No	1952	3.19	0.40	

**Table 9 ijerph-19-03185-t009:** Students’ t-test for home and PMH.

Variable	Out of Home to Go to University	N	M	SD	t (df) p
Factor 1—Personal Satisfaction	Yes	1342	3.20	0.63	−1.56 (3170) 0.120
	No	1830	3.23	0.57	
Factor 2—Prosocial Attitude	Yes	1342	3.58	0.36	0.15 (3170) 0.881
	No	1830	3.57	0.38	
Factor 3—Self-control	Yes	1342	2.75	0.64	−1.99 (3170) 0.047
	No	1830	2.80	0.63	
Factor 4—Autonomy	Yes	1342	3.	0.61	−0.81 (3170) 0.416
	No	1830	3.11	0.57	
Factor 5—Problem-solving and Self-actualization	Yes	1342	3.16	0.50	0.36 (3170) 0.972
	No	1830	3.16	0.49	
Factor 6—Interpersonal Relationship Skills	Yes	1342	3.17	0.50	−1.77 (3170) 0.077
	No	1830	3.21	0.47	
PMH Global	Yes	1342	3.16	0.41	−1.41 (3170) 0.160
	No	1830	3.18	0.39	

## Data Availability

The data presented in this study are available on request from the corresponding author. The data are not publicly available due to ethical considerations.

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
