# Peer review of "Positive Mental Health in University Students and Its Relations with Psychological Vulnerability, Mental Health Literacy, and Sociodemographic Characteristics: A Descriptive Correlational Study"

_ijerph, 2022, doi:10.3390/ijerph19063185_

Round 1

Reviewer 1 Report

My apologies for taking so long to respond; I have been very ill for the last two weeks. Regarding the rewritten manuscript, my concerns have been addressed.

Author Response

Thank you for your review

It was carried out a linguistic revision in all the article according to the declaration in the annex

Reviewer 2 Report

The manuscript is much improved and hypotheses, methods, results, and discussion are all aligned and easy to follow.

I only have one comment about a paragraph in the discussion (lines 336-347). Here, the authors say that their findings show that students who reported more knowledge about PMH were more psychologically vulnerable. Then they report a possible explanation on the positive effects that greater awareness and knowledge have on reducing psychological vulnerability. This seems contradicting, and I would expect a discuss on why greater knowledge may make people more vulnerable. 

Author Response

Thank you for your review

English language and style are fine/minor spell check required. - It was carried out a linguistic revision in all the article according to the declaration in the annex

I only have one comment about a paragraph in the discussion (lines 336-347). Here, the authors say that their findings show that students who reported more knowledge about PMH were more psychologically vulnerable. Then they report a possible explanation on the positive effects that greater awareness and knowledge have on reducing psychological vulnerability. This seems contradicting, and I would expect a discuss on why greater knowledge may make people more vulnerable. - We have reviewed the discussion of this association (lines 339-344).

This manuscript is a resubmission of an earlier submission. The following is a list of the peer review reports and author responses from that submission.

Round 1

Reviewer 1 Report

Review of “Positive mental health in university students and its relations with psychological vulnerability, mental health literacy, gender, residence and scholarship”

14 Dec 2021

This article describes a study of university students’ mental health and relationships to other demographic and psychological factors.

The research looks good, and I have just a few comments to help improve the manuscript.

One of the keywords is “nursing”, but I think this should be removed since nursing is not the only discipline that was surveyed.

In line 58 I would add reference 4, since this is where a reader can find information about the model.

Line 106 mentions STROBE and this needs to be explained. (I was not familiar with the term.)

Line 110—was it 10 months or 11 months? November to September would be 10 months.

Table 2: the numerical categories are reversed, I believe. In line 139 high level is numbers 39-78.

I believe tables 5 and 6 are reversed.

One of the big changes that should be made is that the results do not match up with the six hypotheses. There are no results shown for a correlation between PMH and literacy or for PMH and vulnerability.

Either in the discussion section, or earlier in the introduction, when you talk about scholarships, could you please explain what scholarship means in your institution? For instance, is it only for low-income students, or is it for high-achieving students? This will help readers make sense of the results.

One major limitation that is not mentioned is that the data collection period crossed into the pandemic, so you may have some data pre-pandemic and some during. This needs to be mentioned. (I think it is better to keep the large sample size rather than separate out based on time.)

In the conclusion, nursing students are mentioned again, but they were not the only population studied, so that should be removed.

Author Response

I send the answers to the reviewer as an attachment.

Reviewer 2 Report

The authors conducted a cross-sectional study to assess positive mental health and its relationship with sociodemographic characteristics, mental health literacy, and psychological vulnerability among Portuguese university students. In this study males reported higher personal satisfaction and autonomy compared to females, students not receiving a scholarship reported higher overall positive mental health, self-control, autonomy, and interpersonal relationship skills than those receiving a scholarship, and students who did not change residence showed higher self-control than students who did. Moreover, a significant positive correlation between mental health literacy and both age and psychological vulnerability was found.

The authors focus on positive mental health, which is an important construct that has often been neglected in favor of a view of mental health as simply the absence of psychological distress. While this study highlights needs related to positive mental health in university students and could inform future interventions, there are numerous issues that need to be addressed.

Overall, the paper needs major editing for English language and lacks important details, as described below:

  • The title should include information on the type of study being presented. Moreover, it is difficult to understand what the word “residence” means in the context of this study.
  • The abstract lacks important details such as the type of students included in this study and which course they are enrolled in. In addition, the results are vaguely presented. For example, instead of reporting differences in areas of positive mental health (e.g., between males and females, with and without scholarship, etc.), the authors reported “Statistically significant differences were found between students’ gender, displaced people and scholarship holders concerning some PMH factors” (lines 34-36). I also suggest reporting in parentheses the coefficients and p-values for the main findings.
  • The keywords do not reflect the constructs being analyzed in this study (e.g., mental health literacy, psychological vulnerability, etc.) or the sample of interest (e.g., university students). At the same time, they include words like “nursing”, which is not indicated to be a focus of the study.
  • The introduction does not provide any rationale for assessing mental health literacy and psychological vulnerability, and it does not explain why it is important to focus on positive mental health and university students. The authors should explain how mental health has been considered in other studies (e.g., was mental health considered according to different models of positive mental health or was it considered as absence of distress?) (line 76). References should also be provided for the multifactorial model of positive mental health and the positive mental health questionnaire. Finally, the hypotheses reported at lines 100-102 are vague and do not include any sense of direction.
  • In materials and methods, the authors should specify how the presence of psychiatric pathology was assessed (line 113), explain what the t-test was used for (line 177), and provide references for all questionnaires (lines 127-133).
  • In the results section, in reference to table 3, saying that “Of the students with a low level of PMH, 90.5% were women” (line 200) is not remarkable because this is related to the actual number of participants (e.g., the high levels of females across the entire sample), so reporting the percentage of sex within each level of positive mental health is not informative. What would be more informative is the percentage of positive mental health within each sex. The same concept applies to residence and scholarship. The authors should also explain how the association between positive mental health and gender, residence, and scholarship was calculated (lines 200-201), and they should be consistent when reporting results, making sure to provide coefficients and p-values for all variables. Finally, tables 5 and 6 have been switched in the text.
  • In the discussion, the authors should address the wide age range of their sample and how this may impact the interpretation of results. For example, older students may be more likely to be married, have families and children, be working, etc., and their experience of being a university student may be different than that of younger students.

Author Response

(The authors gave the same response as above.)

Round 2

Reviewer 2 Report

The authors addressed all the comments in a systematic way. However, there are still several issues that need to be addressed.

Abstract:

  • As per the journal’s guidelines, the abstract should be no more than about 200 words. Therefore, the current abstract needs to be significantly shortened.
  • I asked to specify the course students were enrolled in because keywords and conclusions were previously referring to nursing students, while this was not reported in the rest of the manuscript. However, since this has been changed in the revised version of the manuscript, it is not necessary to provide a detailed list of the courses in the abstract (although I would keep this information in the methods section of the manuscript). In the abstract, I suggest deleting the sentence at lines 32-34 “the participants were university students enrolled in different programmes (i.e., health, education, social sciences, economics, engineering, tourism and technology and management).” and adding the word “university” at line 28 as in “in Portuguese university students”.
  • The abstract should include a summary of the main findings. Currently, the abstract includes great detail of numerous findings. For example, the authors report on the correlation between mental health literacy and age, but this is not part of the main study hypotheses. Moreover, details like the coding of each factor of PMH (i.e., F3, F4, etc.) are not necessary and create confusion for readers that are not familiar with the questionnaire being used.
  • At lines 38-42, it is not clear which variables and differences the authors are referring to.For example, are the authors saying that there were significant differences in levels of PMH based on gender, personal satisfaction, autonomy, place of habitual home and scholarship OR that there were significant differences in levels of personal satisfaction and autonomy based on gender, and in PMH factors based on place of habitual home and scholarship?
  • Also, the way the results are being reported in the abstract is not informative. Instead of saying “Statistically significant differences were found between the students’ gender and…” it would be better to specify what these differences are (e.g., “X showed higher/lower levels of Y compared with Z, etc.).
  • PVS needs to be spelled out the first time it is used.

Methods:

  • Chi-square test should be reported in the data analysis section as one of the analyses used. The authors should also provide a detailed description of how they performed this analysis and for which purpose. For example, was PMH considered as a categorical variable (Good, Moderate and Low)?

Results:

  • Overall, it is difficult to match the results with the hypotheses and statistical analyses implemented. For example, both the results for t-test and Chi-square are now being presented, and it is not clear with which of these analyses the authors intend to respond to the last three hypotheses (line 138-141). Is the goal of this paper to analyze the difference on binary categories (i.e., gender, scholarship, and home) in levels of PMH using a t-test or to analyze the association between, for example, gender, scholarship, and home, and categories of PMH using a Chi-square?
  • More results are being presented than necessary to respond to the study hypotheses and goals. The authors should distinguish between primary and secondary outcomes.
  • The results of the Chi-Square should be reported. The authors report r values when presenting Chi-Square results, so it is not clear which results are being presented.

I would also like to point out a few minor points:

  • In the title, the word “residence” was replaced by “home”. I recognize it may be particularly difficult to give a sense of what residence or home mean in the context of a short title. I suggest replacing “gender, place of habitual home and scholarship” with “socio-demographic characteristics” to overcome this issue. Further detail can be given in the main text.
  • The sentence at line 239 needs to be revised, as “basic level education (grades 1 to 4)” seems to be misplaced.
  • At line 163, I suggest replacing “in a unique moment” with “at a single time-point” as it is more idiomatic.
  • There is a mistake in the percentage reported at line 248, where based on the table it should be 67.1% of female university students instead of 69.7%.

Author Response

Thank you very much for the suggestions for improvement.
Sending in the attached document all the changes made. English revision was carried out.

Round 3

Reviewer 2 Report

The authors made the requested revisions. The manuscript has been improved and aims, hypotheses, analyses, and results have been presented more clearly.

The authors proposed to exclude the Chi-Square test results from the paper because of lack of significant findings and because a t-test has been performed. The Chi-square test and the t-test can be used to respond to different research questions, and I think it is important that the authors report the results according to their original statistical plan. I believe the Chi-square test was not reported in the first version of the manuscript, was then added in the second version, and has now been deleted again. I think it is very important to clarify what analyses were intended to be performed to test the study hypotheses, and that results are reported accordingly, whether they are significant or not.

Furthermore, the discussion should reflect the changes made in the rest of the manuscript and address each main finding more systematically:

  • Line 302, the sentence “In addition, the results were not significantly different between genders” seems unrelated to the point being made. The paragraph is about the prevalence of PMH not about gender differences, which is discussed afterwards. Here I would have expected a comment on the differences and similarities found between this study and other studies assessing PMH.
  • Lines 310 and 313, the authors talk about lack of significant associations but it is not clear which associations they are referring to.
  • I suggest combining the second point with the sixth point and elaborating further: were these differences observed in other studies? What possible explanations for the differences in PMH between males and females? Why the authors expected to see better PMH in females and how do they explain the fact that they observed an opposite trend?
  • Line 312, was having regular support assessed? It seems that the study assessed whether students lived at home or not, but not the level of support received. I think the authors may hypothesize that receiving regular support may be a possible explanation why living at home is associated with better PMH.
  • The third point of the discussion should be combined with the paragraph at lines 333-339.
  • I think the fact that PMH is associated with higher mental health literacy, and higher mental health literacy is associated with greater psychological vulnerability should be discussed as one point.
  • Lines 355-360, it is not clear that the findings of this study emphasize the need for free smartphone apps as this has never been mentioned before. Also, the study shows relatively high PMH overall but identify vulnerable groups. I think the authors should reframe this paragraph to highlight this point.

I would also like to highlight a few minor points:

  • Lines 76-77, this sentence seems to not be in line with the rest of the paragraph. It seems like the goal of this paragraph is to list the studies and areas in which positive mental health has been assessed according to the Multifactorial Model, not on studies that focus on racial issues.
  • Line 94, a comma is missing after “residence”. I also notices an Oxford comma is often used but this is not consistent throughout the manuscript. I suggest choosing one style and keep it consistent.
  • Line 129, the months have been removed. Considering that the study covers a period of time that goes from pre Covid-19 pandemic to the midst of it, I think it is important to specify the timeframe ofthe evaluation including the months.
  • Line 159, there is a leftover parenthesis.
  • Line 261, I think the H4 should be reported after table 6 as it does not pertain to the other aspects described in this table, like habitual home and scholarship.
  • Line 263, “and” should be replaced by a comma after students.
  • In the results, when referring to factors of PMH, sometimes F is in parenthesis and sometimes is not. I suggest using a consistent style.